# Generalized Category Discovery with Hierarchical Label Smoothing

## Abstract

*Generalized Category Discovery* seeks to cluster unidentified categories while simultaneously discerning known categories. Existing approaches predominantly rely on contrastive learning to produce distinctive embeddings for both labeled and unlabeled data. Yet, these techniques often suffer from dispersed clusters for unknown categories due to the lack of discriminative cues and a high rate of false negatives, thereby compromising the model's ability to discriminate clusters effectively. To alleviate this problem, we introduce label smoothing as a hyperparameter that permits 'forgivable mistakes' when samples are closely related. We introduce a self-supervised cluster hierarchy, which allows us to control the strength of label smoothing to apply. By assigning pseudo labels to emerging cluster candidates and using these as 'soft supervision' for contrastive learning, we effectively combine the benefits of clustering-based learning and contrastive learning. The resulting method is applicable for both unsupervised and semi-supervised scenarios and we demonstrate state-of-the-art generalized category discovery performance on various fine-grained datasets.

## 1 Introduction

As we continue to advance in computational capacity, abundant labeled datasets, and robust probabilistic models, supervised learning continues to outperform human capabilities in many tasks, for example classifying images within predefined categories He et al. (2016); Krizhevsky et al. (2017); Simonyan & Zisserman (2015); Huang et al. (2017); Szegedy et al. (2015). Despite their superior performance in familiar, well-defined settings, deep learning models fail when faced with the unknown, demonstrating a noticeable shortfall in generalization. This generalization gap can exist on various levels of abstraction. In its most tangible form, this challenge manifests as generalizing to unfamiliar environments featuring the same categories and tasks - a hurdle extensively explored in the fields of *domain adaptation* and *domain generalization* Muandet et al. (2013); Qiao et al. (2020); Zhao et al. (2020); Chuah et al. (2022); Wan et al. (2022); Peng et al. (2022); Zhou et al. (2021). At its most abstract form, this deficit can be seen when models face the need to generalize to previously unseen tasks Lin et al. (2022); Sanh et al. (2022).

Addressing the issue of novel categories can be construed as a middle ground between the two extreme paradigms of known vs. unknown. There are several approaches to this complex issue; a primary strategy could involve discarding any novel category – in essence, bifurcating the data into known versus novel clusters while focusing on the classification of known categories. This strategy is the well-studied problem of *open set recognition* Bendale & Boult (2015; 2016); Scheirer et al. (2012); Vaze et al. (2022a). An alternative methodology involves the transfer of classification knowledge from known to novel categories. This strategy, known as *Novel class discovery* Han et al. (2019); Fini et al. (2021); Xie et al. (2016); Han et al. (2020); Zhao & Han (2021); Zhong et al. (2021b); Joseph et al. (2022); Roy et al. (2022); Rizve et al. (2022); Zhao & Han (2021), aims to categorize new categories, drawing on the classification understanding gained from known categories. Nevertheless, novel class discovery harbors an inherent limitation: the presumption of mutual exclusivity between known and novel categories.

A more pragmatic approach, and the focus of our work, would involve uncovering novel categories concurrently with known ones. To address this concern, the concept of *Generalized Category Discovery* Pu et al. (2023); Zhang et al. (2023); Hao et al. (2023); Chiaroni et al. (2022); Wen et al.

(2022); An et al. (2022) has recently been introduced. The generalized category discovery problem, which provides a model with unlabeled data from both novel and known categories, can be framed as an instance of semi-supervised learning, a well-explored area in machine learning Ouali et al. (2020); Yang et al. (2022); Rebuffi et al. (2020); Oliver et al. (2018); Chapelle et al. (2009). However, the uniqueness of the generalized category discovery lies in the presence of categories without any labeled instances, adding another layer of complexity.

A widely accepted method to tackle semi-supervised learning challenges has been self-supervision Jaiswal et al. (2020); Zhai et al. (2019); Liu et al. (2021), and it is often interchangeably used with the former. Interestingly, contrastive learning Chen et al. (2020); Li et al. (2020); Noroozi & Favaro (2016), a form of self-supervision, has demonstrated its potential in unearthing new semantics, suggesting an enticing path forward in the realm of generalized category discovery Caron et al. (2018); Cao et al. (2022); Han et al. (2020). Despite recent advancements in adopting contrastive learning, a significant challenge persists due to the high occurrence of false negatives within the same category, which adversely impacts semantic clustering quality Caron et al. (2018); Khorasgani et al. (2022); Huynh et al. (2022). The presence of a few false negatives from a category could result in the creation of a fragmented cluster for that category or, even more critically, the misassignment of samples to different clusters. For known category samples, supervised contrastive learning Khosla et al. (2020); Vaze et al. (2022b) could potentially alleviate this issue for known categories with labeled data, thereby enhancing the discriminative capacity of the representation. Nonetheless, the challenge remains due to the absence of supervisory signals for novel categories. Recent studies, such as Zhang et al. (2023); Pu et al. (2023), have attempted to address this issue through the utilization of intermediate concepts as a pseudo-supervisory mechanism. However, given their dependence on similarity graphs, these methods are computationally demanding, thereby limiting their applicability to a broad range of categories.

In this paper, we propose a unique approach that harnesses the power of contrastive learning and pseudo-labeling, avoiding the need for additional data structures such as graphs and, instead, utilizing the model's own ability to employ pseudo-labels hierarchically. We operate on various hierarchy-level representations to discern those that are more illustrative of the input or more discriminative, thereby inclining toward labels. Moreover, we employ the known labels at each stage of the hierarchy, providing more abstract supervision to the representation, which proves advantageous for recognized and novel categories. Ultimately, we utilize the labels generated at each level as pseudo labels for the supervised contrastive learning of unlabeled data.

Our key contributions are as follows:

- Our research leverages hierarchical contrastive learning across varying degrees of supervision, providing us with the unique capability to adjust our dependence on labeled known data and unlabeled unknown data.

- Rather than employing clustering at every level, our approach utilizes the cluster and category centers to generate more abstract labels. This strategy effectively mitigates the heavy computational expense associated with traditional clustering methods, particularly when dealing with a substantial number of categories.

- Empirically, we demonstrate that our novel methodology facilitates effective category discovery and outperforms the existing baselines. Owing to its model-agnostic nature, our approach can be applied to other methods underpinned by contrastive learning, thereby enriching their representational capacity.

## 2 BACKGROUND

The *Generalized Category Discovery* problem introduced by Vaze et al. Vaze et al. (2022b) tries to categorize a set of images during inference, which can be from the known categories seen during training or novel categories. Formally, we only have access to $\mathcal{Y}_{\mathcal{S}}$ or seen categories during training time, while we aim to categorize samples from novel categories or $\mathcal{Y}_{\mathcal{U}}$ during test time. For the Novel Class Discovery problem, it is assumed that $\mathcal{Y}_{\mathcal{S}} \cap \mathcal{Y}_{\mathcal{U}} = \emptyset$. However, this assumption could be unrealistic for real-world data. So Vaze et al. (2022b) proposed to use instead the more realistic Generalized Category Discovery assumption in which the model can encounter both seen and unseen

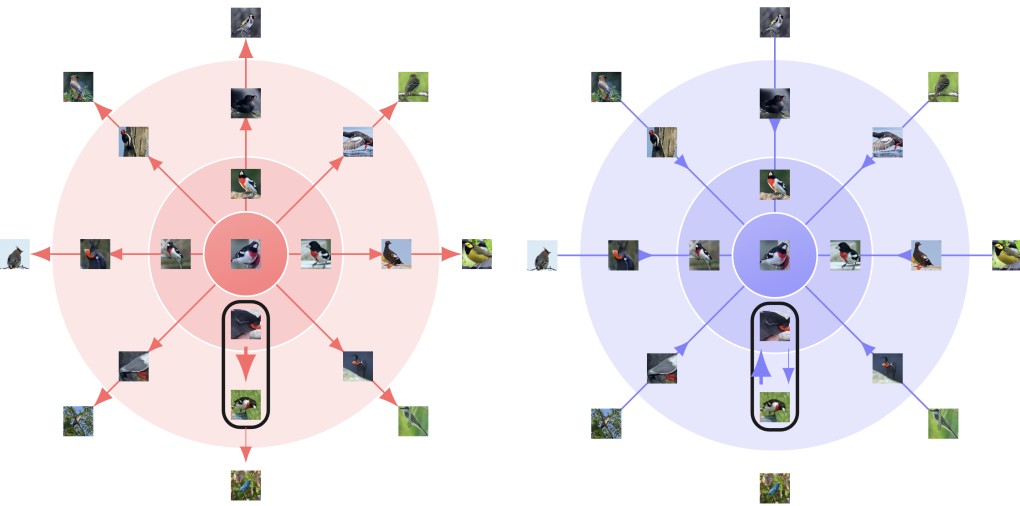

(a) Unsupervised Hierarchical Contrastive Learning  (b) Supervised Hierarchical Contrastive Learning

Figure 1: The motivation for hierarchical unsupervised and supervised learning **(a) Unsupervised Hierarchical Contrastive Learning.** In contrast to conventional unsupervised contrastive learning, our method implicitly divides the feature space into different zones with different strengths of repelling negative samples based on their distance to the positive sample. For instance, in Figure (a), for two samples in the box, our unsupervised contrastive loss repels the samples in lighter shades less strongly than the closer ones. **(b) Supervised Hierarchical Contrastive Learning.** Our supervised contrastive learning also divides feature space into different semantic zones. If two samples are more similar semantically, even if they belong to different categories, the pull between them is stronger. In the box indicated in both Figure (a) and (b), the synergy between these two losses repels the semantically different samples while attracts the semantically similar ones.

categories during test time. In short, for the Generalized Category Discovery problem, we have $\mathcal{Y}_\mathcal{S} \subset \mathcal{Y}_\mathcal{U}$.

The methodology proposed by Vaze et al. (2022b) applies semi-supervised contrastive learning to generate a discriminative and informative embedding for both known and unknown categories. Initially, the method employs unsupervised contrastive learning to discern an image from a multitude of others. However, a noted limitation of unsupervised contrastive learning is its propensity to impose sometimes extreme augmentations as positive instances while treating visually similar samples as negatives, thereby driving the model to increase the distance between similar samples. To circumvent this issue, Vaze et al. (2022b) integrates supervised contrastive learning into their approach, enabling the model to exploit the similarities among samples within the same class. This technique, however, is only applicable to known categories. Enhancing the model's reliance on supervised contrastive learning can improve the clustering quality for known categories at the expense of diminishing the performance of unknown category clusters and vice versa. Extreme reliance on either learning approach compromises the cluster purity for known and unknown categories.

## 3 A PROBABILISTIC APPROACH TO CATEGORY HIERARCHIES

In this section, we provide insights into why hierarchical contrastive learning can provide more information about unseen categories. For contrastive supervised training, the goal is to optimize parameter $\theta$ in order to have the following equation:

$$p_\theta(y = 1|\hat{x}_i, \hat{x}_j) = \delta_{ij} \tag{1}$$

in which $\delta_{ij}$ is the delta Kronecker function which is one only when $i = j$ and zero otherwise. In reality, what we aim for the model to learn through this objective function is the equality of a hidden context variable. Hence, we can consider the following equation:

$$p_\theta(y = 1|c_i, c_j) = \delta_{c_i c_j}. \tag{2}$$

Let's consider the simple Bayesian network depicted in Figure 2. From this diagram, we can calculate the probability of $p(y|\hat{x}_i, \hat{x}_j)$ based on this Bayesian network.

$$p_\theta(y = 1|\hat{x}_i, \hat{x}_j) = \frac{\sum_{x_i} \sum_{x_j} \sum_{c_i} \sum_{c_j} p(y = 1|c_i, c_j)p(c_i)p(c_j)p(x_i|c_i)p(x_j|c_j)p(\hat{x}_i|x_i)p(\hat{x}_j|x_j)}{\sum_{x_i} \sum_{x_j} \sum_{c_i} \sum_{c_j} \sum_y p(y|c_i, c_j)p(c_i)p(c_j)p(x_i|c_i)p(x_j|c_j)p(\hat{x}_i|x_i)p(\hat{x}_j|x_j)}. \tag{3}$$

With some straightforward algebra we can simplify this equation to

$$p_\theta(y = 1|\hat{x}_i, \hat{x}_j) = \sum_{c_i} p(c_i|\hat{x}_i)p(c_i|\hat{x}_j). \tag{4}$$

For $x_i$, we can assume that $\hat{x}_i$ would be a hypersphere with the radius $r_{aug}$. The hypersphere for cluster $c_i$ will have a radius of $R_i$. We can then approximate $p(c_i|\hat{x}_i)$ with the shared volume of hypersphere $c_i$ and hypersphere $\hat{x}_i$. While this can be approximated by the shared cap volume between these two hyperspheres, we can adopt some strategies for simplifying this approximation.

**Strategy one: sample distance zero one**, for this strategy we simply consider that if $x_i$ and $x_j$ do not belong to the same cluster, then $p(c_i|\hat{x}_i)p(c_i|\hat{x}_j)$ is negligible. So, for this strategy, we only consider the negative and positive samples that belong to the same cluster.

**Strategy two: sample distance pairwise**, for this strategy, we consider that $x_i$ and $x_j$ probability of being members of the same cluster will be a function of their pairwise distance.

**Strategy three: cluster distance**, for this strategy, instead of considering the distance to the actual data point $x_j$, we consider the distance of $x_i$ to the center of the cluster containing $x_j$. This strategy can be seen as a combination of the previous two strategies.

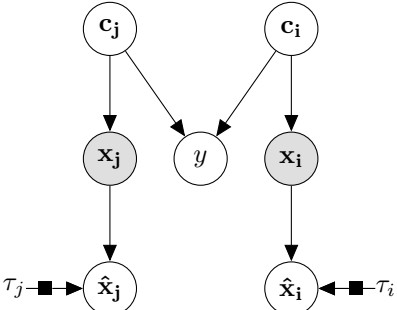

Figure 2: **Baysian Network for the Contrastive Learning Problem.** The shaded nodes are observed variables $x_i$ and $x_j$ which corresponds to the images $i$ and $j$. The augmentations $\hat{x}_i$ and $\hat{x}_j$ created by adding a noise with parameters $\tau_i$ and $\tau_j$ to the actual images.

Now let's consider the label smoothing (Szegedy et al., 2016; Müller et al., 2019) problem for the underlying ground truth distribution, which we will show with $p_\mathcal{T}$. We will have:

$$p_\theta(y = 1|\hat{x}_i, \hat{x}_j) = p_\mathcal{T}(y = 1|c_i, c_j)(1 - \alpha) + \alpha U, \tag{5}$$

$U$ is the uniform distribution over all clusters, which denotes the uncertainty about the ground truth $y$. As we showed with the previous three strategies, we can approximate this $\alpha$ with one of the discussed distances.

### 3.1 Hierarchiacal Contrastive Learning

One thing to consider in equation 2 is that there is no specification about what $c_i$ and $c_j$ could be. They could be as fine as data samples and their augmentations or as abstract as seen vs. unseen categories. We can exploit this property to change the level of abstraction for different categories. This is the foundation for our hierarchical Supervised and unsupervised contrastive learning, which we introduce in the next section. Having defined our theoretical objective, we are now ready to make it operational.

## 4 Hierarchial Contrastive Generalized Category Discovery

In this section, we propose hierarchical unsupervised and supervised contrastive learning to achieve a highly dense and discriminative cluster, which can benefit from the semantic similarities in real-world data. Similar to Vaze et al. (2022b), we use semi-supervised contrastive learning for extracting

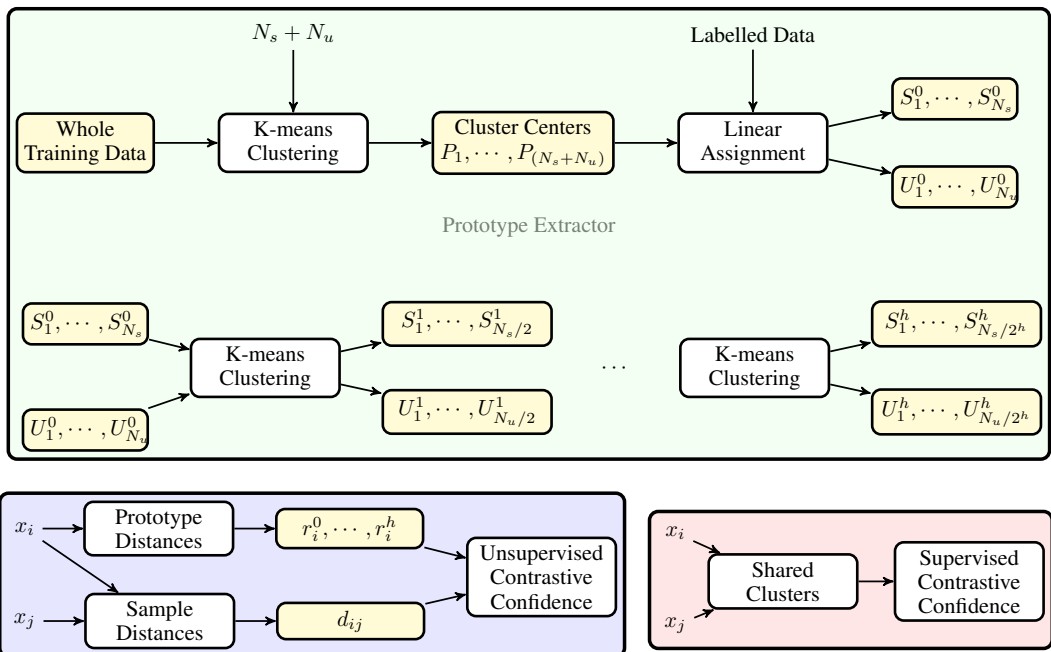

Figure 3: **Proposed framework for hierarchical contrastive learning.** Green box: pipeline for extracting hierarchical prototypes from the input. We then use these prototypes in an hierarchical manner to find confidence in supervised and unsupervised contrastive learning. Blue box: pipeline for our hierarchical unsupervised contrastive learning. For each sample, the method finds all distances from its corresponding hierarchy cluster centers, which we call cluster radiuses. Then, if the distance of sample $x_j$ is higher than each of these radiuses, it decreases its contrastive confidence for this sample. Red box: pipeline for supervised hierarchical contrastive learning. For each hierarchy level, the method considers if $x_i$ and $x_j$ belong to the same cluster in this level.

informative yet discriminative embeddings. One level of semi-supervised contrastive learning could be biased to be more informative by relying more on unsupervised contrastive learning or discriminative by relying on supervised contrastive learning. Instead, to preserve the informative nature of the embedding while enhancing its discriminative properties, our approach deploys various levels of dependency on supervision in a hierarchical fashion, as depicted in Figure 3. The top level of this hierarchy denotes more abstract inclination, as seen vs. unseen, while the lower levels show a finer emphasis as sample vs. sample.

Our hierarchical contrastive learning consists of three phases: pseudo-label extraction, unsupervised, and supervised hierarchical contrastive learning, which we will explain respectively.

**Phase I - Pseudo-label extraction:** To optimize the use of direct supervisory signals and simultaneously avoid over-allocation of the unknown category samples to known category groups, our approach introduces pseudo-labels for unlabelled samples while preserving the ground truth samples for known categories. With each iteration, these pseudo-labels are derived through a combination of clustering and linear assignment. As shown in Figure 3, we first apply clustering to the whole data to extract $N_s + N_u$ clusters and their corresponding centers $P_1, \cdots, P_{N_s+N_u}$ where $N_s$ is the number of seen or known categories and $N_u$ is the number of unknown categories. Given we have the labels for the labeled data, using linear assignment, we divide these cluster prototypes into $N_s$ seen clusters $S_1^0, \cdots, S_N^0$, and $N_u$ unknown clusters $U_1^0, \cdots, U_N^0$. Here, the superindex 0 shows that this is the 0th level of abstraction. Now, for any $i$th level of abstraction, we cluster the prototypes of $i-1$th seen abstraction level into half the amount seen categories and simultaneously $i-1$th unknown prototypes into the half amount. This ensures the prevention of over-assignment of samples to known category clusters and a few select unknown category clusters. Hence, we can anticipate reliable pseudo-labels for different levels of category abstractions.

**Phase II- Unsupervised Hierarchical Contrastive Learning:** Despite the advantageous nature of pseudo-labeling for forming clusters, it presents a significant challenge. In the early stages of training,

the model lacks proficiency in known and unknown labels. Consequently, these pseudo-labels may be inherently noisy, leading to inaccurate model supervision. One can employ unsupervised contrastive learning to use ground truth data to form clusters to counteract these limitations. However, as we discussed before, unsupervised contrastive learning thrives on distinguishing the augmented version of a sample from other samples, including the ones that share a semantic context with the current sample. To counteract these limitations, we employ label smoothing to manage unreliable labels. We also progressively consider more abstract categories; hence, fewer negative samples are discarded for these bigger clusters. But we also progressively decrease the importance of these "bigger cluster" negative samples. As we mentioned in section 3, there are three strategies that we can adopt to find the uncertainty estimation. Two of these strategies rely on the distance from the clusters. However, as we mentioned, there are multiple levels of abstraction and clusters. Hence, if we use strategies other than pairwise distance, we will have the following equation for the label smoothing coefficient $c_l$:

$$c_l = \alpha \sum_{i=0}^{max(lgN_s, lgN_u)} \frac{1}{2^i} D^i \tag{6}$$

in which $lgN_s$ is the logarithm base two of $N_s$, $\alpha$ is the label smoothing hyperparameter, and $D^i$ is the distance metric at the abstraction level $i$. Then, the unsupervised contrastive loss will be:

$$\mathcal{L}_u^{total} = \mathcal{L}_u(c_l) \tag{7}$$

in which $\mathcal{L}_u$ is the conventional unsupervised contrastive learning and $c_l$ is its label smoothing matrix coefficient.

**Phase III- Supervised Hierarchical Contrastive Learning:** The resulting pseudo-labels from the pseudo-labels extraction phase serve as ground truth for the subsequent epoch. So, let us assume $\mathcal{L}_s^i$ is the supervised contrastive learning for the level of the pseudo label $i$. The overall supervised contrastive learning can be calculated as

$$\mathcal{L}_s^{total} = \sum_{i=0}^{max(lgN_s, lgN_u)} \mathcal{L}_s^i. \tag{8}$$

in which $lgN_s$ is the logarithm base two of $N_s$.

Given the availability of labeled samples from known categories, we gradually reduce the impact of label smoothing for known categories as the model becomes more adept at distinguishing them. This strategy ensures that the model generates highly accurate labels for known categories by the end of the training while still delivering valuable pseudo-labels for unknown categories. Since we can access different levels of abstraction, we use supervised contrastive learning for samples belonging to the same cluster. Note that since supervised contrastive learning only considers the samples that are members of the same category or cluster, but it is not important for it to know the exact label, we can adopt this to our advantage by considering the samples that belong to the same cluster,

Finally, our total loss will be:

$$\mathcal{L}^{total} = \mathcal{L}_s^{total} + \mathcal{L}_u^{total} \tag{9}$$

## 5 EXPERIMENTS

### 5.1 EXPERIMENTAL SETUP

**Eight Datasets.** We evaluate our model on three generic datasets CIFAR10 (Krizhevsky et al., 2009), CIFAR100 (Krizhevsky et al., 2009) and ImageNet-100 (Deng et al., 2009) and four fine-grained datasets: CUB (Wah et al., 2011), Aircraft (Maji et al., 2013), Stanford-Cars (Krause et al., 2013) and Oxford-IIIT Pet (Parkhi et al., 2012). Finally, we use the challenging Herbarium19 dataset (Tan et al., 2019), which is fine-grained and long-tailed. Following (Vaze et al., 2022b), we subsample the training dataset in a ratio of $50\%$ of known categories at the train and all samples of unknown categories. For all datasets except CIFAR100, we consider $50\%$ of categories as known categories at training time, while for CIFAR100, $80\%$ of the categories are known during training time. A summary of dataset statistics and their train test splits is shown in Table 1.

**Implementation Details.** Following Vaze et al. (2022b), we use ViT-B/16 as our backbone, which is pre-trained by DINO Caron et al. (2021) on unlabelled ImageNet 1K Krizhevsky et al. (2017), we use

Table 1: **Statistics of datasets and their data splits for the generalized category discovery task.** The first three datasets are generic image classification datasets, while the next four are fine-grained datasets. The Herbarium19 dataset is both fine-grained and long-tailed.

| | Labelled | | Unlabelled | |
|---|---|---|---|---|
| **Dataset** | #Images | #Categories | #Images | #Categories |
| CIFAR-10 (Krizhevsky et al., 2009) | 12.5K | 5 | 37.5K | 10 |
| CIFAR-100 (Krizhevsky et al., 2009) | 20.0K | 80 | 30.0K | 100 |
| ImageNet-100 (Deng et al., 2009) | 31.9K | 50 | 95.3K | 100 |
| CUB-200 (Wah et al., 2011) | 1.5K | 100 | 4.5K | 200 |
| Stanford-Cars (Krause et al., 2013) | 2.0K | 98 | 6.1K | 196 |
| FGVC-Aircraft (Maji et al., 2013) | 3.4K | 50 | 6.6K | 100 |
| Oxford-Pet (Parkhi et al., 2012) | 0.9K | 19 | 2.7K | 37 |
| Herbarium19 (Tan et al., 2019) | 8.9K | 341 | 25.4K | 683 |

the batch size of 128 for training and use $\lambda=0.35$. For label smoothing we use the $\alpha=0.5$. Differnt from Vaze et al. (2022b), we froze 10 blocks of ViT-B/16 and fine-tuned the last two blocks instead of only the last block. The code will be released.

**Evaluation Metrics.** Similar to Vaze et al. (2022b), we use semi-supervised $k$-means to cluster the predicted embeddings. Then, we use the Hungarian algorithm Wright (1990) to solve the optimal assignment of emerged clusters to their ground truth labels. We report the accuracy of the model's predictions on *All*, *Known*, and *Novel* categories. Accuracy on *All* is calculated using the whole unlabelled test set, consisting of known and unknown categories. For *Known*, we only consider the samples with labels known during training. Finally, for *Novel*, we consider samples from the unlabelled categories at train time.

## 5.2 ABLATIVE STUDIES

To ablate the effectiveness of each component independently, we consider different losses of the proposed method. Note that hierarchical unsupervised contrastive learning is defined when we use cluster-based strategies to approximate label smoothing. We consider these two components together.

**Effect of each component.** Table 2 examines the effect of different loss components on the model's overall performance. Since the baseline already uses unsupervised and supervised contrastive learning, in table 2, for unsupervised contrastive learning, we consider its addition, which benefits from hierarchical label smoothing. For supervised contrastive learning, we consider the version without any pseudo-labels as a baseline, while for our loss, we consider its hierarchical supervision using pseudo-labels. Ultimately, when all losses are combined, their synergetic effect helps perform well for both known and novel categories.

**The function of distance used for confidence.** As we mentioned in the method section, the smoothing and uncertainty of the contrastive learning correlate with its distance from the negative sample. There are multiple strategies and distances that we can consider. Sample 0-1 distance only considers the negative samples in the cluster, and for every negative sample beyond the cluster, it will assign the maximum uncertainty possible through the label smoothing hyperparameter. Sample Pairwise Distance: consider the uncertainty or smoothing coefficient as the relative distance of the negative sample to other negative ones in the batch. Finally, sample cluster distance considers the relative distance of the containing cluster of negative samples from the anchor.

**Effects of smoothing hyperparameter** In addition to using sample distances to control uncertainty, we used a smoothing hyperparameter $\alpha$, which controls for the maximum level of uncertainty for the negative samples. For instance, $alpha=0.5$ shows that the model is equally likely to be wrong about a negative sample. While we can use different constants for this hyperparameter, we can also change it towards the end of training to enforce more certain decisions about the negative examples. One strategy would be to change this hyperparameter based on the remaining epoch fraction. This way, there is less room for guessing at the end of the training model, and it should be certain about its assessment. Another strategy is to change this hyperparameter based on the accuracy of labeled data. The intuition about this strategy is that when the model has more mistakes, it should also be less certain about its assessment. These ablations are shown in Table 2.

Table 2: **Ablative studies on the effectiveness of each component.** Accuracy score on the CUB dataset is reported. This table indicates each component's preference for novel or old categories. Their combination enables the model to be robust for both settings. The best settings are bolded.

| Effect of each component | | | |
| --- | --- | --- | --- |
| Method | All | Known | Novel |
| Baseline (Vaze et al., 2022b) | 65.6 | 75.1 | 60.8 |
| Unsupervised w/ Label Smoothing | 64.6 | 76.7 | 58.5 |
| Supervised Hierarchical | 60.9 | 76.7 | 53.0 |
| Combination | **70.4** | **80.3** | **65.4** |

| The distance strategy to be used | | | |
| --- | --- | --- | --- |
| Method | All | Known | Novel |
| Baseline (Vaze et al., 2022b) | 65.6 | 75.1 | 60.8 |
| Sample 0-1 Distance | **70.4** | **80.3** | **65.4** |
| Sample Pairwise Distance | 67.7 | 79.9 | 61.6 |
| Sample Cluster Distance | 69.0 | 79.3 | 63.9 |

| Smoothing Hyperparameter Change | | | |
| --- | --- | --- | --- |
| Method | All | Known | Novel |
| Baseline (Vaze et al., 2022b) | 65.6 | 75.1 | 60.8 |
| Constant | 65.0 | 75.8 | 59.6 |
| Change with epoch | **70.4** | **80.3** | **65.4** |

| Smoothing Hyperparameter Constant | | | |
| --- | --- | --- | --- |
| Method | All | Known | Novel |
| Baseline (Vaze et al., 2022b) | 65.6 | 75.1 | 60.8 |
| $\alpha$=0.1 | 64.6 | 78.1 | 57.9 |
| $\alpha$=0.5 | **70.4** | 80.3 | **65.4** |
| $\alpha$=1.0 | 66.4 | **80.9** | 59.1 |

Table 3: **Comparison with state-of-the-art for coarse-grained image classification.** Accuracy score for the three first methods is reported from Vaze et al. (2022b) and for ORCA from Zhang et al. (2023) and the rest are reported from the corresponding work. Bold and underlined numbers show the best and second-best accuracies. Our method has a consistent performance for the three experimental settings (*All*, *Known*, *Novel*). Our method is especially suitable for novel categories in both datasets.

| | CIFAR-10 | | | CIFAR-100 | | | ImageNet-100 | | |
| --- | --- | --- | --- | --- | --- | --- | --- | --- | --- |
| Method | All | Known | Novel | All | Known | Novel | All | Known | Novel |
| k-means (Arthur & Vassilvitskii, 2007) | 83.6 | 85.7 | 82.5 | 52.0 | 52.2 | 50.8 | 72.7 | 75.5 | 71.3 |
| RankStats+ (Han et al., 2020) | 46.8 | 19.2 | 60.5 | 58.2 | 77.6 | 19.3 | 37.1 | 61.6 | 24.8 |
| UNO+ (Fini et al., 2021) | 68.6 | **98.3** | 53.8 | 69.5 | 80.6 | 47.2 | 70.3 | **95.0** | 57.9 |
| ORCA (Cao et al., 2022) | 96.9 | 95.1 | 97.8 | 74.2 | 82.1 | 67.2 | 79.2 | 93.2 | 72.1 |
| GCD (Vaze et al., 2022b) | 91.5 | _97.9_ | 88.2 | 73.0 | 76.2 | 66.5 | 74.1 | 89.8 | 66.3 |
| XCon(Fei et al., 2022) | 96.0 | 97.3 | 95.4 | 74.2 | 81.2 | 60.3 | 77.6 | _93.5_ | 69.7 |
| PromptCAL (Zhang et al., 2023) | **97.9** | 96.6 | **98.5** | **81.2** | _84.2_ | _75.3_ | **83.1** | 92.7 | **78.3** |
| DCCL (Pu et al., 2023) | 96.3 | 96.5 | 96.9 | 75.3 | 76.8 | 70.2 | 80.5 | 90.5 | 76.2 |
| SimGCD (Wen et al., 2022) | _97.1_ | 95.1 | _98.1_ | _80.1_ | 81.2 | **77.8** | _83.0_ | 93.1 | _77.9_ |
| GPC (Zhao et al., 2023) | 90.6 | 97.6 | 87.0 | 75.4 | **84.6** | 60.1 | 75.3 | 93.4 | 66.7 |
| Ours | 96.4 | 96.5 | 96.3 | 77.4 | 80.9 | 70.4 | 77.0 | 90.1 | 70.5 |

## 5.3 COMPARISON WITH STATE-OF-THE-ART

**Coarse-grained image classification.** We evaluate our model on three generic datasets, namely CIFAR10/100 (Krizhevsky et al., 2009) and ImageNet-100Deng et al. (2009). Table 3 compares our results against state-of-the-art generalized category discovery methods. As we can see from this table, our method performs consistently well on both known and novel datasets. Note that since CIFAR10 only has 10 categories and hence four levels of hierarchies in total, our method can not benefit from all its potential. The same observations, to a higher degree, can be seen on CIFAR100. Another point is that the generic datasets do not always have the hierarchy structure of more fine-grained datasets. also, since the distinctions between clusters are more coarse, the benefit of label smoothing for fine-grained is less substantial than the fine-grained datasets. Nevertheless, Table 3 shows our method performs reasonably well on generic datasets.

**Fine-grained image classification.** Fine-grained image datasets are a more realistic approach to the real world and more aligned with a hierarchical perspective on categories. For generic datasets, visual cues aid the model in discerning the novelty of a category. On the contrary, fine-grained datasets require a more nuanced attention to category-specific details. Table 4 summarizes our model's performance on the fine-grained datasets. As we can see from this table, while performing well on generic datasets, our model has more robust and consistent results than other methods for fine-grained datasets. For the challenging long-tailed Herbarium 19 dataset, our method profits especially from its reliance on clustering prototypes instead of samples for the higher levels of abstraction, as it allows the model to distinguish different categories even from a few examples.

Table 4: **Comparison with state-of-the-art for fine-grained image classification.** Accuracy score for the three first methods is reported from Vaze et al. (2022b) and for ORCA from Zhang et al. (2023) and the rest are reported from the corresponding work. Bold and underlined numbers, respectively, show the best and second-best accuracies. Our method has strong performance for the three experimental settings (*All*, *Known*, and *Novel*). This table shows that our method is especially suited for fine-grained settings with many categories.

| Method | CUB-200 | | | FGVC-Aircraft | | | Herbarium-19 | | | Stanford-Cars | | | Oxford-IIIT Pet | | |
|---|---|---|---|---|---|---|---|---|---|---|---|---|---|---|---|
| | All | Known | Novel | All | Known | Novel | All | Known | Novel | All | Known | Novel | All | Known | Novel |
| k-means (Arthur & Vassilvitskii, 2007) | 34.3 | 38.9 | 32.1 | 12.9 | 12.9 | 12.8 | 13.0 | 12.2 | 13.4 | 12.8 | 10.6 | 13.8 | 77.1 | 70.1 | 80.7 |
| RankStats+ (Han et al., 2020) | 33.3 | 51.6 | 24.2 | 26.9 | 36.4 | 22.2 | 27.9 | 55.8 | 12.8 | 28.3 | 61.8 | 12.1 | - | - | - |
| UNO+ (Fini et al., 2021) | 35.1 | 49.0 | 28.1 | 40.3 | 56.4 | 32.2 | 28.3 | 53.7 | 14.7 | 35.5 | 70.5 | 18.6 | - | - | - |
| ORCA (Cao et al., 2022) | 36.3 | 43.8 | 32.6 | 31.6 | 32.0 | 31.4 | 24.6 | 26.5 | 23.7 | 31.9 | 42.2 | 26.9 | - | - | - |
| GCD (Vaze et al., 2022b) | 51.3 | 56.6 | 48.7 | 45.0 | 41.1 | 46.9 | 35.4 | 51.0 | 27.0 | 39.0 | 57.6 | 29.9 | 80.2 | 85.1 | 77.6 |
| XCon (Fei et al., 2022) | 52.1 | 54.3 | 51.0 | 47.7 | 44.4 | 49.4 | - | - | - | 40.5 | 58.8 | 31.7 | 86.7 | **91.5** | 84.1 |
| PromptCAL (Zhang et al., 2023) | 62.9 | 64.4 | 62.1 | 52.2 | 52.2 | _52.3_ | - | - | - | 50.2 | 70.1 | 40.6 | - | - | - |
| DCCL (Pu et al., 2023) | _63.5_ | 60.8 | _64.9_ | - | - | - | - | - | - | 43.1 | 55.7 | 36.2 | **88.1** | 88.2 | **88.0** |
| SimGCD (Wen et al., 2022) | 60.3 | _65.6_ | 57.7 | _54.2_ | _59.1_ | 51.8 | **44.0** | _58.0_ | 36.4 | **53.8** | **71.9** | _45.0_ | - | - | - |
| GPC (Zhao et al., 2023) | 52.0 | 55.5 | 47.5 | 43.3 | 40.7 | 44.8 | - | - | - | 38.2 | 58.9 | 27.4 | - | - | - |
| Ours | **70.4** | **80.3** | **65.4** | **59.8** | **70.6** | **54.5** | _41.2_ | **59.0** | _31.6_ | _51.7_ | 60.8 | **47.2** | _87.4_ | _91.2_ | _85.4_ |

# 6 RELATED WORKS

**Novel Category Discovery** can be traced back to Han et al. (2019), where they used the knowledge from labeled data to infer the unknown categories. Following this work, Zhong et al. (2021a) solidified the novel class discovery as a new specific problem. The main goal of novel class discovery is to transfer the implicit category structure from the known categories to infer unknown categories Fini et al. (2021); Xie et al. (2016); Han et al. (2020); Zhao & Han (2021); Zhong et al. (2021b); Joseph et al. (2022); Roy et al. (2022); Rizve et al. (2022); Zhao & Han (2021). Prior to the novel class discovery, the problem of encountering new classes at the test time was investigated by open-set recognition Bendale & Boult (2015; 2016); Scheirer et al. (2012); Vaze et al. (2022a). However, the strategy of dealing with these new categories is different. In the open-set scenario, the model rejects the samples from novel categories, while novel class discovery aims to benefit from the vast knowledge of the unknown realm and infer the categories. However, the novel class discovery has a limiting assumption that test data only consists of novel categories. For a more realistic setting, *Generalized Category Discovery* considers both known and old categories at the test time.

**Generalised Category Discovery** is introduced by Vaze et al. (2022b) and concurrently under the name *Open-world semi-supervised learning* by Cao et al. (2022). In this scenario, while the model should not lose its grasp on old categories, it must discover novel categories in test time. This adds an extra challenge because when we adapt the novel class discovery methods to this scenario, they try to be biased to either novel or old categories and miss the other group. It has been a recent surge of interest in generalized category discovery Pu et al. (2023); Zhang et al. (2023); Hao et al. (2023); Chiaroni et al. (2022); Wen et al. (2022); An et al. (2022). In this work, instead of viewing categories as an end, we investigated the fundamental question of how to conceptualize *category* itself.

# 7 CONCLUSION

This work leverages hierarchical contrastive learning to discover unknown categories in conjunction with the known ones. Traditional contrastive learning models for unknown categories typically focus on two extremes: the ultimate labels given for known categories and the instance-level contrastive learning for a mix of labeled and unlabeled data. While some attempts have been made to avoid this binary approach through the use of similarity graphs, our study employs varying degrees of hierarchy, from the most unsupervised to the most supervised, to seek out representations. This method enables the model to place varying importance on known and unknown data, and it also establishes intermediary steps for learning these representations. Furthermore, we use clustering and linear assignment to extract pseudo labels for the subsequent supervised contrastive learning. The use of these pseudo-labels facilitates supervised contrastive learning for the unlabeled data, thereby enhancing the training speed and the integrity of the clusters formed for unknown categories. Since these pseudo-labels can also employed at different levels of hierarchies, they provide informative supervision signals for different abstraction levels. Finally, by employing the label smoothing hyperparameter, we let the model adopt unsupervised contrastive learning in a more local scope and focus on finer distinctions. This in the end leads to a stronger fine-grained ability for our model.

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
