# OpenReview forum: "Generalized Category Discovery with Hierarchical Label Smoothing"
_ICLR.cc/2024/Conference — ICLR 2024 Conference Withdrawn Submission_

### Official Review · Reviewer_KPLz · 2023-10-19

**Soundness:** 2 fair
**Presentation:** 2 fair
**Contribution:** 2 fair
**Rating:** 3
**Confidence:** 4

**Summary:**

This paper discusses the challenge of models' ability to generalize to both known and unknown categories, which is vital for real-world applications. The authors propose a novel method to deal with the generalized category discovery problem, which leverages a combination of contrastive learning and pseudo-labeling. They introduce a hierarchical contrastive learning framework that adapts to varying degrees of supervision and abstraction levels, improving the representation of both known and unknown categories.

**Strengths:**

- The research problem is realistic and important. Knowing what it doesn't know is crucial for machine.

- Experimental results are detailed with extensive ablation studies.

**Weaknesses:**

- The presentation of the proposed is unclear and lacks justification. In Phase 1, why the method can ensure "the prevention of over-assignment of samples to known category clusters and a few select unknown category clusters"? How to ensure that the known-category samples are assigned to the appropriate clusters? How to give pseudo-labels? And the losses are also not clear.

- While Figures 1 and 3 occupy substantial real estate within the document, their informative value is notably low. Figure 1's content lacks clarity, leaving readers uncertain about the significance of the varying shades and arrows employed. Furthermore, Figure 3 fails to provide a comprehensive understanding of the algorithmic process.

- While the proposed algorithm shows promise, there are certain aspects of its effectiveness that warrant further discussion. An examination of the results in Table 3 reveals that when compared to existing approaches such as PromptCAL, DCCL, SimGCD, and GPC, it may not exhibit the same level of overall performance. These competing methods seem to outperform the algorithm introduced in this paper.

**Questions:**

For clarity, the description of the main contributions is not sufficiently clear, and the proposed algorithm lacks support, not even on an intuitive level. Based on the current version, I cannot assess the novelty of the paper. Although the studied problem is interesting, the quality of this paper could be further enhanced.

---

### Official Review · Reviewer_x4PR · 2023-10-30

**Soundness:** 2 fair
**Presentation:** 2 fair
**Contribution:** 2 fair
**Rating:** 3
**Confidence:** 4

**Summary:**

This paper is aimed at the task of Generalized Category Discovery, which provides a model with unlabeled data from both novel and known categories. By introducing a self-supervised cluster hierarchy and assigning pseudo labels to emerging cluster candidates for soft supervision in contrastive learning, the proposed method demonstrates a state-of-art performance on various corresponding datasets.

**Strengths:**

1. The proposed approach utilizes hierarchical contrastive learning across varying degrees of supervision, providing the unique capability to adjust the dependence on labeled known data and unlabeled unknown data.
2. During the implementation of the proposed approach, the cluster and category centers are utilized to generate more abstract labels, which alleviates the heavy computational expanse.
3. Experimental results demonstrate the effectiveness of the proposed method.

**Weaknesses:**

1. In Section 2 Background, it is suggested to use more necessary mathematical symbols to clearly explain the task of Generalized Category Discovery in mathematical language. For example, the dataset, the purpose of the algorithm, and the hierarchy, and the used contrastive learning should be introduced formally.
2. Some paragraphs are poorly organized. For example, to be more persuasive, the second paragraph in Section 2 Background and the second paragraph in Section 6 Related Works should be incorporated, due to that the purpose of the content includes the differences between previous works and this work and why the proposed method has advantage.
3. Some necessary notations are missing, which makes the analysis and algorithm confusing. For example, in Eq. (1), the meanings of $y, \hat{x}, p(y=1 \vert \hat{x}_i, \hat{x}_j)$ are unknown. In Eq. (2), the meaning of $c_i$ is unknown.
4. The emphasis of the article is inconsistent. For example, in Abstract, it is mentioned that to alleviate the problem of the previous work, this work introduces label smoothing as a hyper-parameter that permits ‘forgivable mistakes’, which sounds interesting. However, some details about this interesting part are not given in the main body. At least, how it works, why it works, and the relationship with contrastive learning should be explained.
5. The figures are suggested to be polished up. For example, Figure 2 should give more vivid details of the framework instead of some plain texts.

**Questions:**

1. Why does label smoothing help?
2. Is the number of cluster centers unknown in the task?
3. How does the Linear Assignment module work in the proposed framework?

---

### Official Review · Reviewer_2Rzo · 2023-10-30

**Soundness:** 2 fair
**Presentation:** 1 poor
**Contribution:** 1 poor
**Rating:** 3
**Confidence:** 4

**Summary:**

The paper introduces a method for integrating contrastive learning with pseudo-labeling, negating the need for extra data structures like graphs. The approach employs hierarchical representations and known labels to discern the most informative representations. Instead of traditional clustering, it uses cluster and category centers to derive abstract labels, reducing computational demands. Experiments are performed on several benchmark datasets.

**Strengths:**

- Generalized category discovery is a highly significant yet understudied problem in the literature for various reasons. Their effort to elucidate on this subject is commendable and emphasizes the potential value and impact of their work in advancing the domain.

**Weaknesses:**

- The paper resorts to basic and ad-hoc assumptions in an attempt to address a challenging problem. There's an evident disconnect between the heuristic method detailed in Section 4 and the Bayesian belief network concept outlined in Section 3. This lack of coherence and clarity raises concerns about the foundational premises of the proposed solution and its theoretical rigor.

- When juxtaposed with state-of-the-art methodologies, the results produced by the proposed approach seem to fall short of expectations. This raises questions about its efficacy and relevance in the current research landscape, potentially limiting its applicability and acceptance within the community.

- The writing quality in several segments is subpar, leading to ambiguities and making it challenging for readers to grasp and navigate through specific sections. This undermines the accessibility of the content and the paper's potential impact on its intended audience.

**Questions:**

Other comments:


Generalized category discovery problem was introduced and studied long before Vaze et al. and other papers cited in the related work. See papers below.

Miller, D. J., & Browning, J. (2003). A mixture model and EM-based algorithm for class discovery, robust classification, and outlier rejection in mixed labeled/unlabeled data sets. IEEE Transactions on Pattern Analysis and Machine Intelligence, 25(11), 1468-1483.

Dundar, Murat, Feri Akova, Yuan Qi, and Bartek Rajwa. "Bayesian nonexhaustive learning for online discovery and modeling of emerging classes." In Proceedings of the 29th International Coference on Machine Learning, pp. 99-106. 2012.

The method assumes N_u is known. How would one know the number of unseen categories beforehand?

In the following sentence I am not sure "or" makes much sense. It is supposed to be "and".

"We operate on various hierarchy-level representations to discern those that are more illustrative of the input or more discriminative,"

What does y=1 signifies in Equation 1? Please define the notation first before using them.

Equation 4 is missing a "y".

Figure 2 caption has typos.

Many parts of the paper poorly written and difficult to follow. For example, not sure what the "half the amount seen categories" in the following sentence means.

"Now, for any ith level of abstraction, we cluster the prototypes
of i − 1th seen abstraction level into half the amount seen categories and simultaneously i − 1th
unknown prototypes into the half amount."

---

### Official Review · Reviewer_QpG6 · 2023-11-01

**Soundness:** 3 good
**Presentation:** 1 poor
**Contribution:** 1 poor
**Rating:** 3
**Confidence:** 5

**Summary:**

This paper tackles the generalized category discovery problem with a hierarchical contrastive learning apporach.
The method first clusters all the data into clusters with the labelled data fixed to the labelled clusters. Then, more abstract levels are built on top of the first level of clustering.
Further, a label smoothing technique is introducted to learn better representations.
Experiments show improvements on harder SSB benchmarks.

**Strengths:**

1. The proposed method indeed gives performance boost to some extend.
2. The paper gives nice ablation study of the proposed method.

**Weaknesses:**

1. Overall, the usage of hierarchical contrastive learning may not be so novel, for example [R1] already uses the concept of hierarchical contrastive learning to learn transferrable representations, this paper should have discussions on [R1], and justify the use of hierarchical contrastive learning in GCD.
2. Some claims of contribution is not fully supported, for example, 1) in the introduction it is said that the proposed method could mitigates the heavy computational expense of clustering, but no data is given to validate this, maybe adding some comparison on training time with baselines like Vaze etal and SimGCD? 2) It is also said that the proposed method can be applied to other methods underpinned by contrastive learning, but only one method is tested (Vaze etal), adding other models such as SimGCD and DCCL could make the paper stronger.
3. The performance on coarser-grained data falls behind many other methods, and the gain on finer-grained data is not very clear. Perhaps adding the experiment as I have pointed out in weakness 2 could be helpful.
4. The presentation of the paper is not clear, many notations are used without introduced. For example, I don't know what distance metric is used in Eq 6, and what distance is it referring to. Also for the loss functions, no formula is given to show how to compute the label smoothing contrastive learning.


[R1] HCSC: Hierarchical Contrastive Selective Coding, CVPR 2022

**Questions:**

1. What is the relation of the proposed method to [R1]?
2. Please provide results of combining the proposed method with other GCD methods.
3. Please clarify the notations used in the paper.